ⓐ | **Open Peer Review** | Bacteriology | Methods and Protocols
# Performance of time-lapsed turbidimetry and agar plating as bacterial quantification methods

Angelika Dewicki,[1] Matthew Henkel,[1] Norie Sugitani,[1] Alexander Applegate,[1] Brian T. Campfield[1,2]

**ABSTRACT** Accurate bacterial quantification is crucial for studying microbial pathogenesis, host-pathogen interactions, and therapeutic interventions. Two widely used methods include agar plating with colony-forming unit (CFU) enumeration and time-lapsed turbidimetry in liquid broth culture. While agar plating remains the gold standard in both *in vitro* and *in vivo* infection models, liquid broth turbidimetry is commonly used to assess growth kinetics, microbial fitness, antibiotic susceptibility, and bacterial genetics. While strain-specific CFU and turbidimetry (OD) calibration studies exist, a comprehensive and systematic comparison of these methods for quantifying a broad array of clinically relevant pathogens remains largely unexplored. Here, we conducted a head-to-head comparison of agar plating and liquid broth turbidimetry to quantify the growth of *Klebsiella pneumoniae*, *Pseudomonas aeruginosa*, and *Staphylococcus aureus in vitro* and evaluated their performance *in vivo* using a murine model of *K. pneumoniae* pulmonary infection. Across all pathogens tested, both methods exhibited strong correlation over a broad dynamic range (6–7 $\log_{10}$ dilutions). Liquid broth turbidimetry demonstrated enhanced sensitivity at low bacterial densities, as well as greater precision. In the *in vivo* murine pneumonia model, this method more accurately distinguished bacterial burdens at the site of infection (lung) and dissemination (spleen) between wild-type and Toll-like receptor 4 knockout mice. Overall, liquid broth turbidimetry is a reliable alternative to agar plating with a high degree of correlation for bacterial quantification and improved precision, highlighting its potential utility in studies of bacteriology and infectious diseases.

**IMPORTANCE** Accurate bacterial quantification is fundamental to microbiology research and clinical diagnostics. While agar plating is a widely used method, our study demonstrates that liquid broth turbidimetry provides accurate bacterial quantification, which is fundamental to microbiology research and clinical diagnostics. While agar plating is a widely used method, our study demonstrates that liquid broth turbidimetry provides a highly correlative and more precise approach for quantifying diverse, clinically relevant bacterial pathogens, including when dealing with low bacterial density. The enhanced sensitivity and precision may be valuable for early infection detection, monitoring treatment efficacy, and understanding microbial dynamics in research settings where accurate quantification of even low-density organisms is essential. The findings support considering liquid broth turbidimetry as a complementary or alternative method for bacterial growth quantification.

**KEYWORDS** bacterial quantification, bacteriology, mouse model, pneumonia, *Klebsiella pneumoniae*, *Pseudomonas aeruginosa*, *Staphylococcus aureus*

M odeling infection requires precise and accurate methods for quantification of bacterial growth to understand the dynamics of infection, evaluate the efficacy of antimicrobial interventions, and other aspects of basic microbiological research.

**Peer Reviewer** Rafael M. Cantón, Hospital Universitario Ramon y Cajal, Madrid, Spain

Address correspondence to Brian T. Campfield, brian.campfield@chp.edu.

The authors declare no conflict of interest.

See the funding table on p. 12.

Two common methods for quantifying bacterial load are colony-forming unit (CFU) enumeration on agar plates and liquid broth culture time-lapsed spectrophotometry (liquid broth turbidimetry). Agar plating, which enumerates CFUs on a semisolid medium (agar), provides a direct measure of viable bacterial cells and is considered a standard for certain applications (1). However, this method can be susceptible to error due to technical variation resulting from manual handling required for serial dilutions, the sample spreading technique, and the observer-dependent accuracy of manual colony counting on plates with high or low bacterial densities. Liquid broth turbidimetry offers a higher-throughput alternative for assessing bacterial growth kinetics by quantifying optical density over time (2). This approach facilitates real-time monitoring of bacterial growth and enables efficient application for such uses as screening antimicrobial compounds and assessing the genetics of bacterial fitness and growth (3). While both approaches are foundational in bacteriology and provide valuable insights, their relative performance and suitability in different experimental contexts require careful consideration.

*Klebsiella pneumoniae*, *Pseudomonas aeruginosa*, and *Staphylococcus aureus* are common human pathogens of importance (4, 5), manifesting clinically as pneumonia, bacteremia, and urinary tract infections, causes of substantial morbidity and mortality (6). Improved understanding of the pathogenesis and host response to these infections is urgently needed for improved diagnostic and therapeutic strategies (7). While previous comparative studies focused on *in vitro* calibration (8), the comparative performance characteristics of these bacterial quantification methods are a significant gap in the field, which warrants a comprehensive, head-to-head analysis. Moreover, validating the *in vitro* comparative findings using an *in vivo* model can further extend the value of each quantification method.

This study aimed to provide a direct comparison between traditional agar plating and liquid broth turbidimetry methods for quantifying bacterial growth in both *in vitro* and *in vivo* contexts of bacterial pneumonia, using four bacterial strains often utilized in modeling pulmonary infection: *K. pneumoniae* serotype K2 (*Kp43816*), antibiotic-resistant *K. pneumoniae* serotype K1 (*Kp396*) (9), *P. aeruginosa* (*PA01*), and *S. aureus* (*USA300*). The *in vitro* component of this study evaluated the sensitivity and dynamic range of both bacterial quantification methods across the three bacterial species, assessing the correlation between the obtained measurements. Furthermore, *in vivo* investigation of the performance of these methodologies during a murine *K. pneumoniae* lung infection model quantified bacterial growth in lung and spleen tissues from mice of both wild-type (WT) and Toll-like receptor 4 knockout (*Tlr4*$^{-/-}$), a relevant immunodeficient genotype, using both agar plating and time-lapsed turbidimetry techniques. A head-to-head comparison of agar plating and liquid broth turbidimetry was conducted to quantify the bacterial burden of multiple clinically relevant pathogens in both controlled *in vitro* systems and complex *in vivo* murine tissue samples.

## MATERIALS AND METHODS

### Bacterial strains and culture conditions

Four bacterial strains were utilized in this study: *K. pneumoniae* serotype K2 (*Kp43816*) (ATCC), *K. pneumoniae* serotype K1 (*Kp396*) (a gift from the Kolls laboratory) (10), *P. aeruginosa* (*PA01*) (a gift from the Alcorn laboratory), and *S. aureus* (*USA300*) (a gift from the Alcorn laboratory). Each bacterial strain was inoculated from frozen stock into 2 mL tryptic soy broth (TSB) (Remel) and cultured overnight at 37°C under aerobic conditions with shaking at 250 revolutions per minute. Overnight cultures were then used to generate subcultures for the following *in vitro* and *in vivo* experiments as described below.

## Bacterial quantification by agar plating

To determine the viable bacterial count, the CFU assay was performed. For each bacterial strain, a subculture was prepared using 45 µL of overnight culture into 2 mL of TSB for a target concentration of $1 \times 10^9$ CFU/mL. Each subculture was serially diluted in sterile phosphate-buffered saline (PBS) (Gibco) using 10-fold serial dilutions. The dilution series ranged from 1:1 to 1:10,000,000,000 in sterile PBS. 10 µL of each bacterial dilution was then plated onto Luria-Bertani (LB) agar plates made with LB broth (Sigma-Aldrich) using sterile inoculum loops. LB is a standard, rich, non-selective medium employed broadly in microbiological research. The use of a single, standardized medium ensures consistent conditions across the panel of studied pathogens. Plates were incubated at 37°C for 18–24 h to facilitate the formation of discrete bacterial colonies. The number of distinct bacterial colonies on each plate was manually counted, and the CFU per milliliter was calculated using the equation (CFU) × (dilution factor) × (µL total sample volume) / (µL plated). All agar plating was performed in duplicate. The limit of detection for this assay was defined as the greatest dilution at which discernible colonies were observed.

## Bacterial quantification by liquid broth turbidimetry

For each bacterial strain, the same culture specimens prepared for agar plating were utilized and were subjected to 10-fold serial dilutions in sterile TSB medium ranging from 1:1 to 1:10,000,000,000 in a sterile flat-bottom 96-well plate. Each dilution was prepared in technical triplicate in adjacent wells of the plate. Control wells containing only sterile TSB medium were included on each plate to serve as blanks and to monitor for potential contamination.

The 96-well flat-bottom plate was placed into a temperature-controlled plate reader (Synergy H1 Hybrid Reader). The plate was incubated at a constant temperature of 37°C for a total duration of 12 h. During the 12-h incubation, bacterial growth was monitored by measuring the optical density at 600 nm ($OD_{600}$) using the plate reader's spectrophotometer. $OD_{600}$ readings were automatically recorded at 30-minute intervals throughout the incubation. The plate reader was programmed to gently shake the plate continuously throughout the incubation. Growth curves were generated for each dilution and replicate by plotting the $OD_{600}$ values against the corresponding time points in hours. Lag time was determined using the Gen5 data analysis software integrated with the Synergy H1 Hybrid Reader. The software identifies the transition from lag to exponential growth by fitting the kinetic $OD_{600}$ data to a non-linear regression model. The lag time is computed as the time-axis intercept between the initial $OD_{600}$ baseline and the tangent line corresponding to the point of maximum specific growth rate on the exponential phase of the fitted growth curve. Lag time was recorded as the duration (hours) of this phase, determined by the transition to exponential growth.

## Murine pulmonary infection and tissue collection

A comparative experiment was performed of these two bacterial quantification methodologies (agar plate vs liquid broth) *in vivo* employing a murine model of pulmonary infection. WT and $Tlr4^{-/-}$ mice were used. WT C57BL/6J and $Tlr4^{-/-}$ mice (11) were purchased from Jackson Laboratory. $Tlr4$, a critical component of innate immune function, is a canonical receptor for pathogen-associated molecular patterns, specifically lipopolysaccharide, found in gram-negative bacteria. Age- and sex-matched male and female mice between 9 and 12 weeks old were used for this study. Mice were infected under anesthesia with 50 µL of *K. pneumoniae* K2 serotype (*Kp43816*) with a target inoculum of 100 CFU/mL via intratracheal instillation. At 48 h post-infection, mice were euthanized, and lung and spleen tissues were immediately harvested under sterile conditions. Each lung and each spleen were placed into a sterile 5 mL culture tube containing 1 mL sterile PBS and homogenized with PRO Scientific 02-07095 horn homogenizer. The resulting tissue homogenates were then transferred to sterile 1.5 mL tubes.

## Bacterial quantification of infected tissue homogenates via agar plating

Homogenized lung tissue and spleen tissue samples were serially diluted to 1:1, 1:100, 1:1,000, and 1:10,000 in sterile PBS. For each dilution, 10 µL aliquots were spread onto LB agar plates using sterile inoculum loops. Plates were incubated at 37°C for 18–24 h to facilitate the formation of discrete bacterial colonies. Following incubation, the number of bacterial colonies was manually counted, and the CFU per milliliter per tissue was calculated. All plating was performed in duplicate for each dilution. The limit of detection was defined as above.

## Bacterial quantification of infected tissue homogenates via liquid broth turbidimetry

Homogenized lung tissue samples were diluted to 1:100, and homogenized spleen tissue samples were diluted to 1:50 in TSB. These dilutions were completed in duplicate in a 96-well flat-bottom plate. Control wells containing only sterile TSB medium were included on each plate to serve as blanks. The 96-well plate was then incubated in a Synergy H1 Hybrid reader as described above. Growth curves were generated for each tissue sample and replicate by plotting OD values against time.

### Statistical analysis

For the *in vitro* comparison of bacterial quantification methods, a simple linear regression analysis was performed to assess the correlation between the two methods across the 10-fold dilution series for each bacterial isolate. The correlation coefficient ($R^2$) was calculated to quantify the strength and direction of the linear relationship. The percent standard error of the mean (SEM) was calculated for every dilution within each quantification method to assess the variability.

For the *in vivo* experiments assessing bacterial growth at the site of infection (lung) and dissemination (spleen) in WT and $Tlr4^{-/-}$ mice infected with *Kp43816*, statistical significance between the two groups was determined using a two-tailed *t*-test with Welch's correction to account for potential unequal variances between the WT and $Tlr4^{-/-}$ groups. This analysis was performed separately for the agar plating of lung and spleen homogenates, as well as for the liquid broth turbidimetry growth data of lung and spleen homogenates. The resulting *P* values were used to determine statistical significance, with a threshold of $P < 0.05$ considered statistically significant. Additionally, a simple linear regression analysis was performed to assess the correlation between the bacterial growth quantified by agar plating and the liquid broth turbidimetry assessed by time-lapsed spectrophotometry for both lung and spleen tissues. $R^2$ was calculated to evaluate the strength of this correlation *in vivo*. The percent SEM was also calculated for both quantification methods within the *in vivo* sample groups. Statistical analyses were conducted using GraphPad 10.1.1 (Prism).

## RESULTS

### *In vitro* comparison of agar plating and liquid broth turbidimetry

The sensitivity and dynamic range of the agar plating and liquid broth turbidimetry methods were compared with a variety of bacterial strains related to pulmonary infection: *K. pneumoniae* serotype K2 (*Kp43816*), *K. pneumoniae* serotype K1 (*Kp396*) (9, 10, 12), *P. aeruginosa* (*PA01*), and *S. aureus* (*USA300*). For the lactose-fermenting *Kp43816*, LB agar plates yielded quantifiable growth across seven orders of magnitude, from the undiluted sample (1:1) to a 1:1,000,000 dilution. Beyond this dilution, no discernible colony formation was observed (Fig. 1A), establishing $8.1 \times 10^2$ CFU/mL as the limit of detection for this method under these conditions. In contrast, quantification via liquid broth turbidimetry demonstrated detectable bacterial growth at an additional 10-fold dilution to 1:10,000,000 (Fig. 1B), indicating enhanced sensitivity of approximately 1 log (12) unit greater than that achieved with agar plating for *Kp43816*, identifying that the

liquid broth turbidimetry method can detect viable bacteria at lower concentrations than were detectable on a semisolid medium. A strong linear correlation ($R^2$ = 0.9967) was observed between the CFU counts obtained from agar plating and the growth kinetics observed in liquid broth turbidimetry across the quantifiable dilution ranges (Fig. 1C). Notably, the percent SEM was consistently similar or lower for liquid broth turbidimetry compared to agar plating across nearly all comparable dilutions (Fig. 1D), suggesting that the liquid broth turbidimetry offers greater precision in quantifying

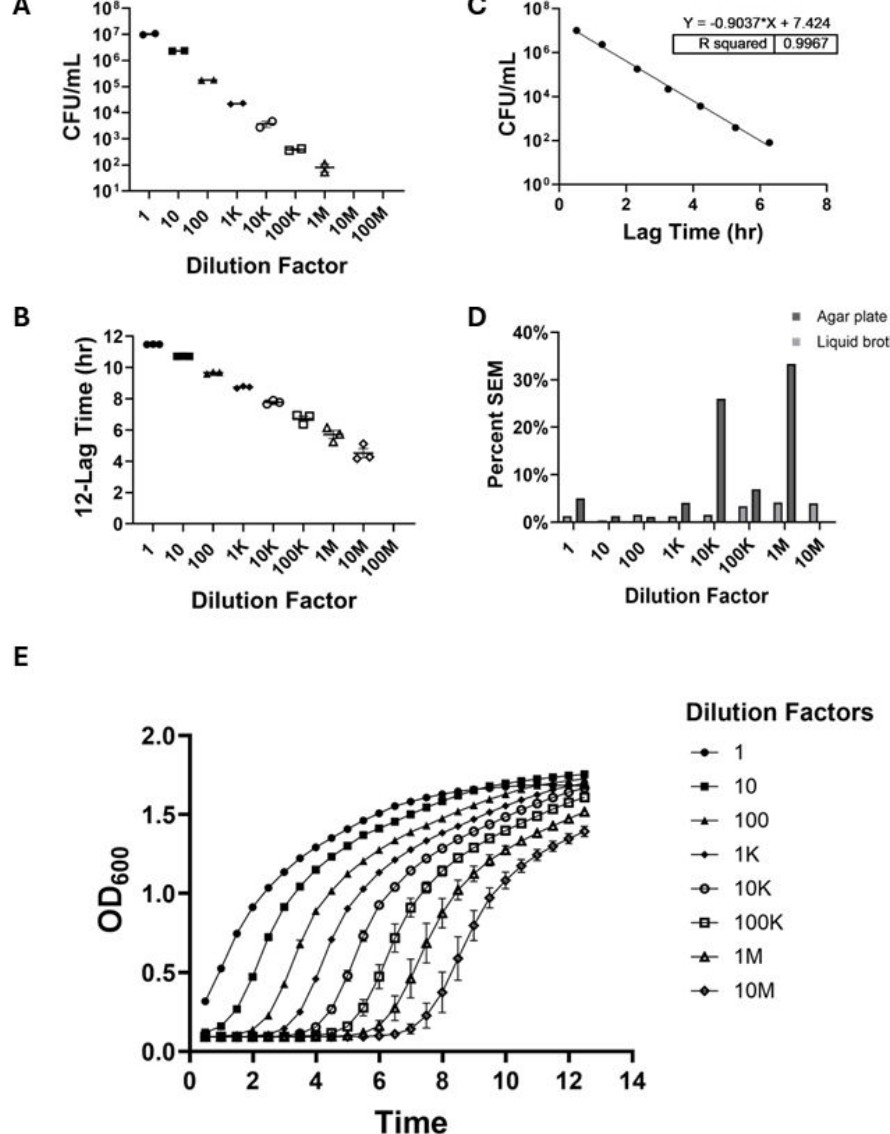

**FIG 1** Comparison of bacterial quantification methods of *Klebsiella pneumoniae* K2 serotype (*Kp43816*). (A) LB agar plate quantification of *Kp43816* by colony-forming units (CFU) at 1:1–1:1,000,000 dilutions (*n* = 2). As expected from the 10-fold serial dilution series, the quantified CFU decreased by a factor of 10 for each successive dilution. No colony growth was observed beyond 1:1,000,000 dilution. (B) Liquid tryptic soy broth quantification of *Kp43816* determined by time-lapsed spectrophotometry via plate reader at 1:1–1:10,000,000 dilutions (*n* = 3). Growth is represented as total run time (12 h) minus the lag time. No growth beyond 1:10,000,000 dilution. As expected from the 10-fold serial dilution series, the quantified lag time decreased by about an hour for each successive dilution. (C) Correlation of agar plates and liquid broth quantification for *Kp43816* cultures in 10-fold dilution series. A simple linear regression analysis was used for comparison ($R^2$ = 0.9967). (D) Percent SEM for agar plates and liquid broth quantification across all dilutions with *Kp43816* growth. (E) Representative growth curves obtained by time-lapsed spectrophotometry via plate reader for *Kp43816* cultures from 1:1 to 1:10,000,000,000 dilutions. K, thousand; M, million.

*Kp43816*. Representative growth curves of *Kp43816* generated from the liquid broth turbidimetry data illustrated the dynamic growth across different dilutions (Fig. 1E).

To examine whether these characteristics were generalizable to other relevant pathogens, analogous experiments were conducted using *Kp396*, *PA01*, and *USA300* (Fig. 2 to 4, respectively), revealing similar trends. To assess a clinical strain of *K. pneumoniae*, we employed the serotype K1 (*Kp396*) hypervirulent (string-test positive) isolate (9, 10, 12), where agar plating yielded quantifiable growth across 6 $\log_{10}$ dilutions with a limit of detection of $1.53 \times 10^2$ CFU/mL (up to a 1:100,000 dilution), while spectrophotometric quantification in liquid broth turbidimetry extended detection 100-fold further to a 1:10,000,000 dilution (Fig. 2A and B). Again, a strong positive correlation ($R^2 = 0.9843$)

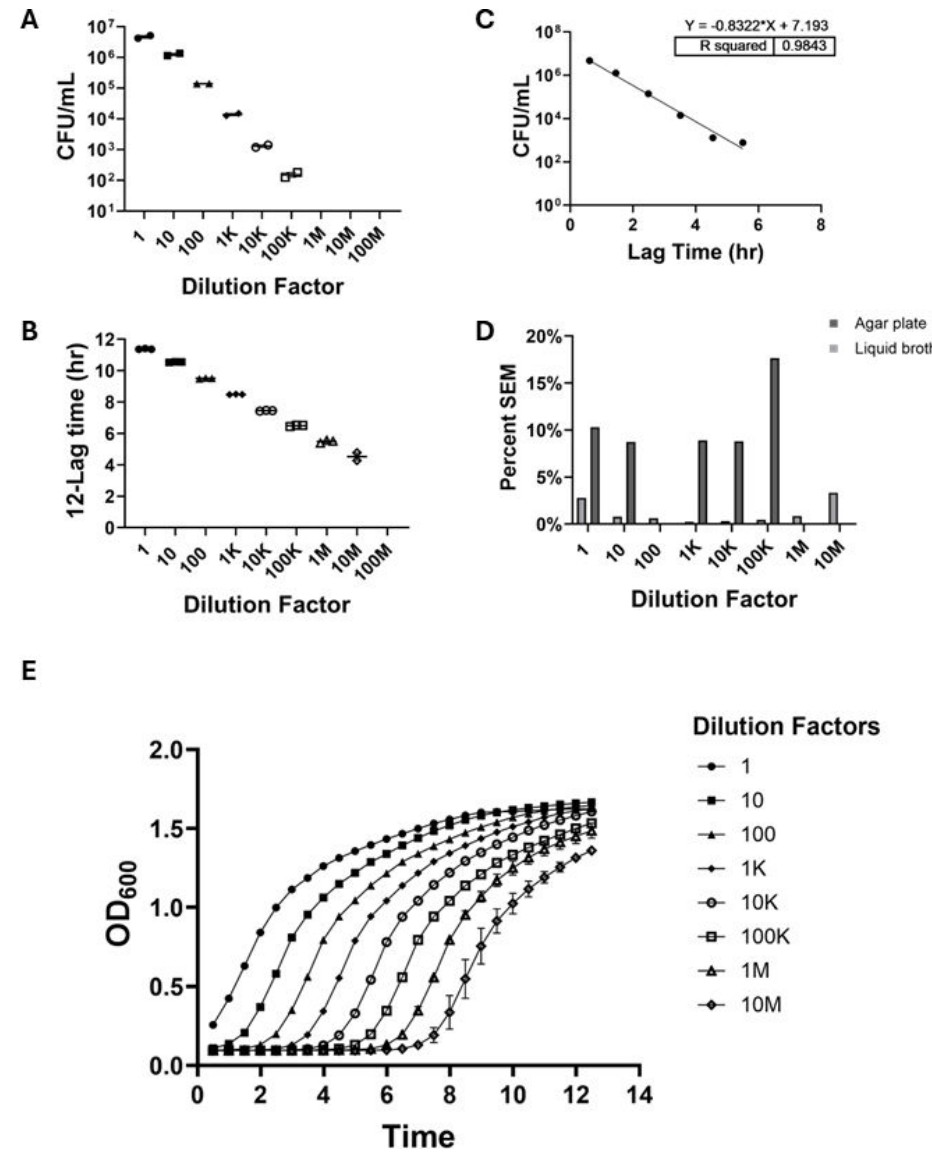

**FIG 2** Comparison of bacterial quantification methods of *Klebsiella pneumoniae* K1 serotype (*Kp396*). (A) LB agar plate quantification of *Kp396* by CFU at 1:1–1:100,000 dilutions (*n* = 2). No colony growth was observed beyond 1:100,000 dilution. (B) Liquid TSB quantification of *Kp396* determined by time-lapsed spectrophotometry via plate reader at 1:1–1:10,000,000 dilutions (*n* = 3). Growth is represented as total run time (12 h) minus the lag time. No growth beyond 1:10,000,000 dilution. (C) Correlation of agar plates and liquid broth quantification for *Kp396* cultures in 10-fold dilution series. A simple linear regression analysis was used for comparison ($R^2 = 0.9843$). (D) Percent SEM for agar plates and liquid broth quantification across all dilutions with *Kp396* growth. (E) Representative growth curves obtained by time-lapsed spectrophotometry via plate reader for *Kp396* cultures from 1:1 to 1:10,000,000,000 dilutions. K, thousand; M, million.

was observed between the two methods in their overlapping quantifiable growth (Fig. 2C), and the percent SEM was consistently lower for the liquid broth turbidimetry across dilutions (Fig. 2D). Representative growth curves for *Kp396* generated from the liquid broth turbidimetry data illustrate the dynamic growth across different dilutions (Fig. 2E).

Next, to compare these results with the non-lactose fermenting *PA01*, agar plating quantified bacterial growth from 1:1 to 1:100,000 dilutions with a limit of detection of $4.5 \times 10^2$ CFU/mL (Fig. 3A), whereas liquid broth turbidimetry extended the quantifiable range to a 1:1,000,000 dilution (Fig. 3B), demonstrating a 10-fold greater sensitivity for the liquid broth turbidimetry method. A high correlation ($R^2 = 0.9949$) was observed between the two methods (Fig. 3C), and the liquid broth turbidimetry method generally exhibited lower percent SEM values (Fig. 3D). Representative growth curves for *PA01* generated from the liquid broth turbidimetry data illustrated the dynamic growth across different dilutions (Fig. 3E).

Finally, to further compare these findings with a gram-positive pathogen, we utilized *USA300*, wherein agar plating allowed for quantification from 1:1 to 1:100,000 dilutions with a limit of detection of $6.3 \times 10^1$ CFU/mL (Fig. 4A), while liquid broth turbidimetry extended the detection limit 10-fold to 1:1,000,000 dilution (Fig. 4B). A strong correlation ($R^2 = 0.9939$) was observed (Fig. 4C), and the liquid broth turbidimetry method showed a trend toward lower percent SEM values across the dilutions (Fig. 4D). Representative growth curves for *USA300* generated from the liquid broth turbidimetry data illustrated the dynamic growth across different dilutions (Fig. 4E). Overall, the data identified that bacterial quantification using agar plating was highly correlative with liquid broth turbidimetry across multiple clinically relevant pathogenic species. Liquid broth turbidimetry was more sensitive than agar plating at detecting growth in more dilute samples, suggesting a capability to detect lower bacterial densities. Additionally, liquid broth turbidimetry quantification had a similar or smaller percent SEM for each species and strain, including in highly diluted samples, consistent with the greater precision of this method. Liquid broth turbidimetry consistently demonstrates a 10-fold increase in sensitivity compared to the agar plating standard.

### *In vivo* comparison of bacterial growth quantification from murine tissue

Having characterized the performance of liquid broth turbidimetry relative to CFU counting with agar plating for assessing bacterial growth *in vitro*, we next sought to determine how these techniques compared in the context of an active infection *in vivo*. To address this, a well-established murine pneumonia model was utilized by infecting WT and *Tlr4$^{-/-}$* (highly susceptible to gram-negative infection) mice with *Kp43816* to induce pulmonary infection. By quantifying *K. pneumoniae* in these tissues from both genotypes, the application of these quantification methods in a clinically relevant *in vivo* host defense model was assessed. Bacterial growth in lung and spleen tissues harvested at 48 h post-infection was quantified using agar plating and liquid broth turbidimetry methods as described above. In lung tissue homogenates, the standard CFU quantification method revealed a modest, statistically non-significant elevation in bacterial burden in *Tlr4$^{-/-}$* compared to WT mice ($4.03832 \times 10^6 \pm 2.40139 \times 10^6$ SEM CFU/mL for WT vs $1.1494 \times 10^8 \pm 5.0163 \times 10^7$ SEM CFU/mL for *Tlr4$^{-/-}$*; $P = 0.0915$; Fig. 5A). When the same lung homogenates were analyzed using liquid broth turbidimetry, the difference in bacterial burden was statistically significant, with *Tlr4$^{-/-}$* mice demonstrating higher burden compared to WT mice (mean of $6.988 \pm 0.511$ SEM h for WT vs $8.846 \pm 0.195$ SEM h for *Tlr4$^{-/-}$*; $P = 0.0185$; Fig. 5C). Bacterial dissemination was also determined to be greater in *Tlr4$^{-/-}$* mice compared to WT as determined in spleen tissue homogenates. Here, both methods detected a statistically significant difference between *Tlr4$^{-/-}$* and WT mice. The level of significance was greater with the liquid broth turbidimetry (mean of $6.027 \pm 0.338$ SEM h for WT vs mean $8.885 \pm 0.223$ SEM h for *Tlr4*; $P = 0.0002$; Fig. 5D) than with the agar plating method ($6.6740 \times 10^4 \pm 4.2016 \times 10^4$ SEM CFU/mL for WT vs $7.6360 \times 10^7 \pm 2.4075 \times 10^7$ SEM CFU/mL for *Tlr4$^{-/-}$*; $P = 0.0339$; Fig. 5B). A strong linear correlation was observed between agar plating and turbidimetric quantification in liquid

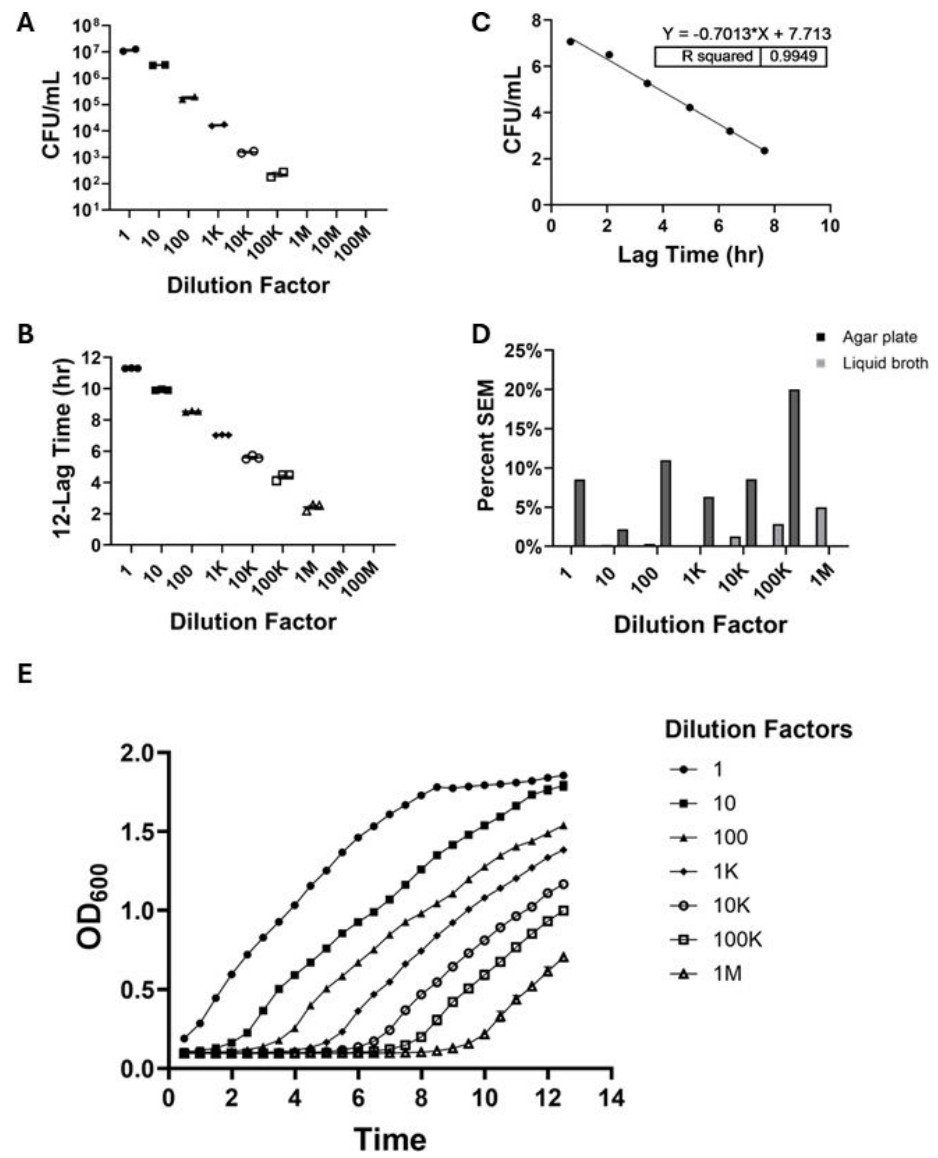

**FIG 3** Comparison of bacterial quantification methods of *Pseudomonas aeruginosa* (*PA01*). (A) LB agar plate quantification of *PA01* by CFU at 1:1–1:100,000 dilutions (*n* = 2). No colony growth was observed beyond 1:100,000 dilution. (B) Liquid TSB quantification of *PA01* determined by time-lapsed spectrophotometry via plate reader at 1:1–1:1,000,000 dilutions (*n* = 3). Growth is represented as the total run time (12 h) minus the lag time. No growth beyond 1:1,000,000 dilution. (C) Correlation of agar plates and liquid broth quantification for *PA01* cultures in 10-fold dilution series. A simple linear regression analysis was used for comparison ($R^2$ = 0.9949). (D) Percent SEM for agar plates and liquid broth quantification across all dilutions with *PA01* growth. (E) Representative growth curves obtained by time-lapsed spectrophotometry via plate reader for *PA01* cultures from 1:1 to 1:1,000,000,000 dilutions. K, thousand; M, million.

broth cultures for both lung ($R^2$ = 0.9318) and spleen ($R^2$ = 0.9796) tissues (Fig. 5E and F). This correlation further validates the liquid broth turbidimetry assay as a reliable method for assessing bacterial growth *in vivo*, providing comparable quantitative information to traditional agar plating and CFU counting. The percent SEM of the mean was significantly improved for the liquid broth turbidimetry method compared to the agar plate culture method (Fig. 5G and H).

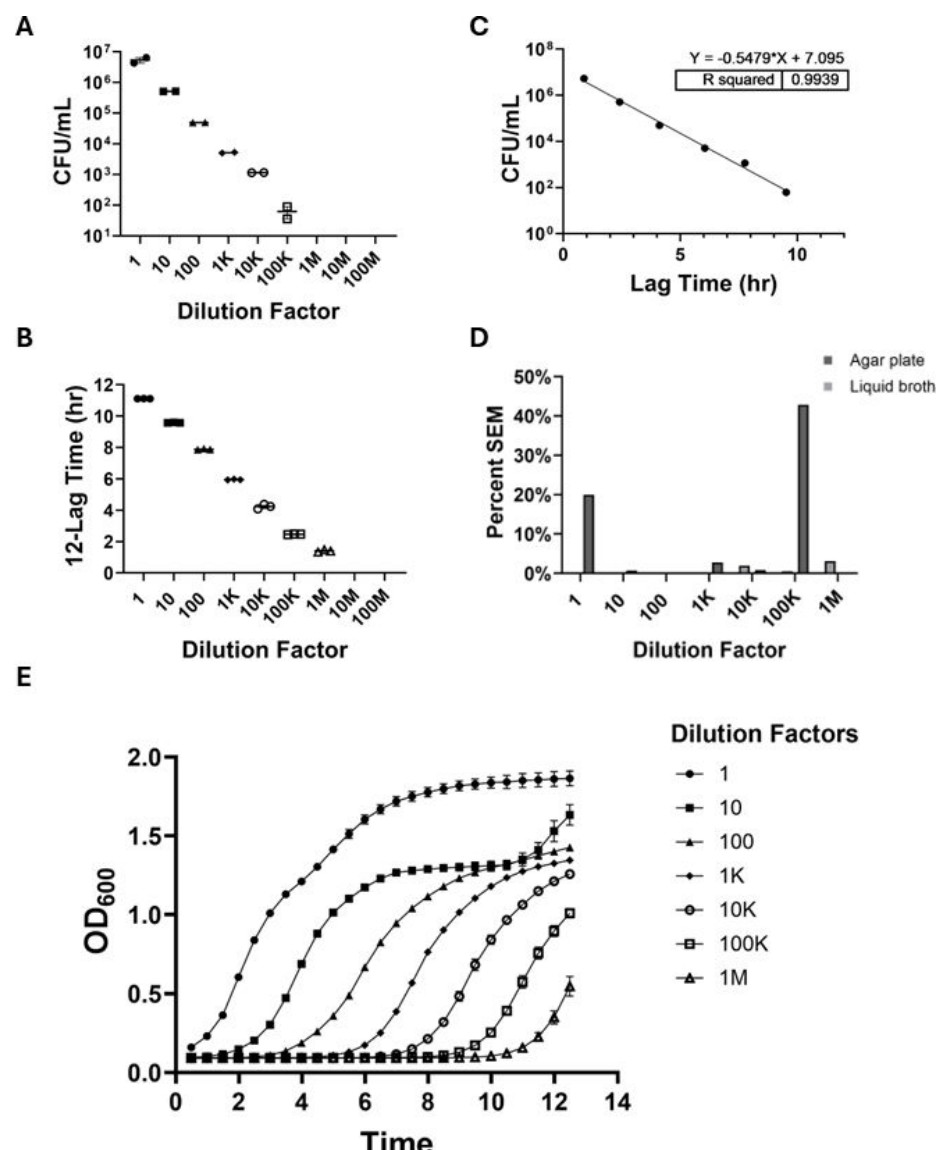

**FIG 4** Comparison of bacterial quantification methods for *Staphylococcus aureus* (*USA300*). (A) LB agar plate quantification of *USA300* by CFU at 1:1–1:100,000 dilutions (*n* = 2). No colony growth was observed beyond 1:100,000 dilution. (B) Liquid TSB quantification of *USA300* determined by time-lapsed spectrophotometry via plate reader at 1:1–1:1,000,000 dilutions (*n* = 3). Growth is represented as the total run time (12 h) minus the lag time. No growth beyond 1:1,000,000 dilution. (C) Correlation of agar plates and liquid broth quantification for *USA300* cultures in 10-fold dilution series. A simple linear regression analysis was used for comparison ($R^2$ = 0.9939). (D) Percent standard SEM for agar plate and liquid broth quantification across all dilutions with *USA300* growth. (E) Representative growth curves obtained by time-lapsed spectrophotometry via plate reader for *USA300* cultures from 1:1 to 1:1,000,000,000 dilutions. K, thousand; M, million.

## DISCUSSION

This study provides a comparative analysis of bacterial quantification using traditional agar plating and time-lapsed spectrophotometry in liquid broth culture (turbidimetry) across both *in vitro* and *in vivo* experimental models. While both methods provided comparable results for quantifying the growth of *K. pneumoniae* (serotype K2 and K1), *P. aeruginosa*, and *S. aureus in vitro*, liquid broth turbidimetry consistently demonstrated a superior limit of detection and greater precision. The strong linear correlations observed between the two methods within the overlapping quantifiable ranges underscore the value of liquid broth turbidimetry as a reliable and quantitatively comparable alternative

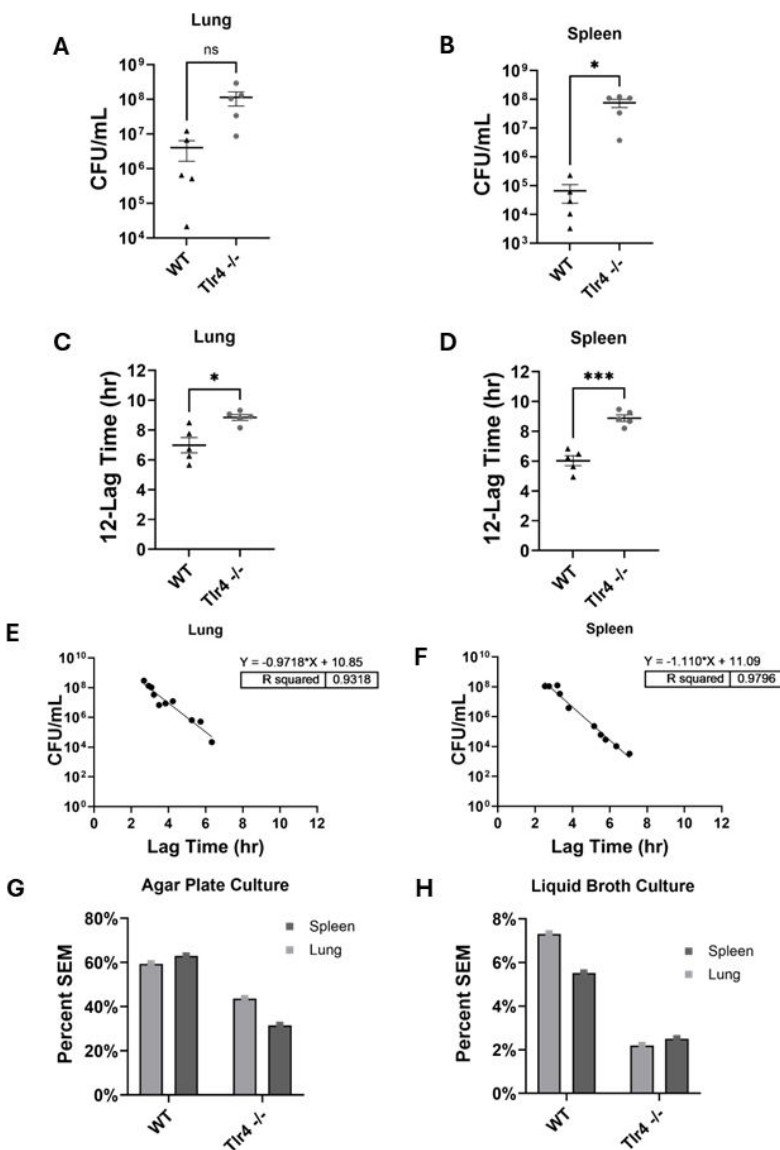

**FIG 5** Comparison of bacterial quantification methods from murine tissue following *K. pneumoniae* (*Kp43816*) pulmonary infection *in vivo*. Representative data from three separate trials. (A) Lung bacterial burden quantification by agar plate method for *Kp43816* infection in wild-type (WT) and Toll-like receptor 4 knockout (*Tlr4*$^{-/-}$) mice (*n* = 5/group). Statistical significance was determined using a two-tailed *t*-test with Welch's correction (*P* = 0.0915). (B) Spleen bacterial burden quantification by agar plate (CFU) for *Kp43816* infection in WT and *Tlr4*$^{-/-}$ mice (*n* = 5/group). Statistical significance was determined using a two-tailed *t*-test with Welch's correction (*P* = 0.0339). (C) Lung bacterial burden quantification determined by liquid broth method for *Kp43816* infection in WT and *Tlr4*$^{-/-}$ mice (*n* = 5/group). Growth is represented as the total run time (12 h) minus the lag time. Statistical significance was determined using a two-tailed *t*-test with Welch's correction (*P* = 0.0185). (D) Spleen lag time determined using a plate reader for *Kp43816* infection in WT and *Tlr4*$^{-/-}$ mice (*n* = 5/group). Growth is represented as the total run time (12 h) minus the lag time. Statistical significance was determined using a two-tailed *t*-test with Welch's correction (*P* = 0.0002). (E and F) Correlation of agar plate and liquid broth quantification for lung (E) and spleen (F) bacterial burden. A simple linear regression analysis was used for comparison for lung (*R*$^2$ = 0.9318) and spleen (*R*$^2$ = 0.9796). (G and H) Percent SEM was greater for agar plating (G) compared with liquid broth (H) quantification of lung and spleen samples from WT and *Tlr4*$^{-/-}$ agar plate mice infected with *Kp43816*. \**P* < 0.05; \*\*\**P* < 0.001; ns, not significant.

for assessing bacterial growth dynamics and bacterial load under controlled laboratory conditions. The enhanced sensitivity enables the detection of bacterial growth at lower concentrations, holding promise for applications involving low-abundance samples, the study of early stages of infection, or in the evaluation of antimicrobial treatment response, where the goal is to rapidly reduce the viable bacterial burden. Furthermore, the consistently lower percent SEM associated with the liquid broth turbidimetry shows a higher degree of precision, likely attributable to the automated and continuous nature of the measurements, which minimizes the inherent manual handling variability associated with agar plating.

The *in vivo* component of this study, using a murine model of pulmonary *K. pneumoniae* infection in WT and immunodeficient *Tlr4*$^{-/-}$ mice, yielded critical insights into the relative sensitivity of these quantification methods within a complex biological system. While agar plating detected a significant increase in splenic bacterial burden in *Tlr4*$^{-/-}$ mice, consistent with impaired systemic clearance, it failed to resolve differences in the lung, likely due to limited precision. In contrast, liquid broth turbidimetry revealed statistically significant differences in both lung and spleen, reflecting its greater sensitivity and precision, suggesting that the liquid broth turbidimetry can discern subtle but biologically significant variations in bacterial load within heterogenous tissue samples. In the context of gram-negative lung infection, *Tlr4* is an essential host defense receptor, and the magnitude of the lung bacterial burden is directly linked to disease severity, systemic dissemination, and mortality in *K. pneumoniae* mouse models (9, 13, 14). The increase in statistical power thus enables a more nuanced understanding of *in vivo* infection dynamics.

Several factors likely contribute to the increased sensitivity and reliability of liquid broth turbidimetry, particularly in the analysis of *in vivo* samples. The process of tissue homogenization and subsequent serial dilution inherent to agar plating introduces potential variability and may lead to an underestimation of the true viable bacterial count due to uneven distribution within the homogenate. Conversely, the liquid broth turbidimetry allows for the incubation of a larger initial sample volume in a nutrient-rich environment, potentially enhancing the growth from low numbers of viable bacteria that may not form distinct colonies on agar within the standard incubation period.

The use of time-lapsed turbidimetry relies on the established principle that the duration of the lag phase is inversely proportional to the initial viable bacterial cell concentration in a culture. Consequently, the greater the number of cells introduced to the fresh medium, the sooner the population enters the exponential growth phase, reducing the measurable lag time (15). The strong linear correlations across the diverse range of pathogens and complex *in vivo* samples (Fig. 1 to 5) provide empirical validation for applying this established relationship to quantifying bacterial burden in experimental settings.

Despite the strong correlations observed between the two methods *in vivo*, liquid broth turbidimetry was more sensitive at detecting different bacterial quantities in tissue samples between WT and *Tlr4*$^{-/-}$ mice, demonstrating successful application of liquid broth turbidimetry to *in vivo* infected tissue samples. Importantly, the significantly lower percent SEM values associated with the liquid broth method in *in vivo* samples consistently showed improved precision and reduced variability when quantifying bacterial load in complex biological systems, which are inherently more variable than *in vitro* cultures.

Although the culture conditions described here are effective for all organisms tested, factors such as shaking speed, oxygen availability, and incubation settings may require adjustment for other bacteria (15–20). A key limitation of the study's scope was that aerobic organisms were utilized; a systematic comparison of the two quantification methods was not attempted for obligate anaerobes due to their requirement of a specialized plate reader capable of supporting strict anaerobic conditions for the turbidimetry analysis. Furthermore, the use of a 10 µL plating volume for agar plating CFU per milliliter quantification may have established higher limits of detection,

which preclude quantification of very low bacterial concentrations present in samples. Finally, while liquid broth turbidimetry offers high sensitivity, a significant barrier to its widespread adoption in clinical practice is due to CFU quantification being a deeply entrenched standard in clinical infectious disease medicine, rendering the clinical application of liquid broth turbidimetry a substantial change in clinical practice. It is worth noting that the initial investment for a high-throughput, shaking plate reader and compatible software represents a significantly greater expense than the consumables required for traditional agar plating, a factor that may limit widespread adaptation in resource-limited settings. Future work should explore alternative culture conditions or methods to address these challenges.

These findings build upon previous work demonstrating the utility of plate-based assays (3) and provide evidence for the advantages of liquid broth culture time-lapsed spectrophotometry over agar plating for accurate and sensitive quantification of bacterial growth. The enhanced sensitivity and reliability of the liquid broth method offers a potentially advantageous technique for quantifying infection dynamics and evaluating the efficacy of therapeutic interventions in preclinical studies, especially when subtle, biological differences in bacterial loads must be reliably discerned. This leads to a more refined understanding of host-pathogen interactions. Expanding this methodology to a broader range of pathogens and infection models beyond pulmonary diseases would further validate its versatility. Nonetheless, the performance of liquid broth turbidimetry makes it a valuable tool for *in vitro* and *in vivo* bacteriology and infection research, particularly where subtle differences in bacterial burden can have significant biological insights.

## ACKNOWLEDGMENTS

This research was supported by the National Institutes of Health (grant R01HL158576, awarded to B.T.C.). The authors thank Dr. Jay Kolls for the *K. pneumoniae* (Kp396) isolate and Dr. John F. Alcorn for providing the *P. aeruginosa* (PA01) and *S. aureus* (USA300) isolates.

## AUTHOR AFFILIATIONS

[1]Division of Pediatric Infectious Diseases, Department of Pediatrics, University of Pittsburgh School of Medicine, Pittsburgh, Pennsylvania, USA
[2]University of Pittsburgh Medical Center (UPMC) Children's Hospital of Pittsburgh, Pittsburgh, Pennsylvania, USA

## AUTHOR ORCIDs

Angelika Dewicki http://orcid.org/0000-0002-8335-4538
Brian T. Campfield http://orcid.org/0000-0002-5380-9837

## FUNDING

| Funder | Grant(s) | Author(s) |
| --- | --- | --- |
| National Heart, Lung, and Blood Institute | 1R01HL158576-01 | Brian T. Campfield |

## AUTHOR CONTRIBUTIONS

Angelika Dewicki, Conceptualization, Formal analysis, Investigation, Methodology, Validation, Visualization, Writing – original draft, Writing – review and editing | Matthew Henkel, Conceptualization, Formal analysis, Investigation, Methodology, Writing – review and editing | Norie Sugitani, Conceptualization, Formal analysis, Investigation, Methodology, Writing – review and editing | Alexander Applegate, Conceptualization, Methodology, Writing – review and editing | Brian T. Campfield, Conceptualization, Formal analysis, Funding acquisition, Methodology, Project administration, Resources, Supervision, Visualization, Writing – review and editing

## ETHICS APPROVAL

All animal protocols were approved by the University of Pittsburgh Institutional Animal Care and Use Committee (protocols #22061470 and #23104015), and all experiments were conducted in accordance with the guidelines and regulations set forth in the Animal Welfare Act and PHS Policy on Humane Care and Use of Laboratory Animals.

## ADDITIONAL FILES

The following material is available online.

### Open Peer Review

**PEER REVIEW HISTORY (review-history.pdf).** An accounting of the reviewer comments and feedback.

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
