## [Reviewer comments · Microbiology Spectrum]

Microbiology Spectrum

Performance of Time-Lapsed Turbidimetry and Agar Plating as Bacterial Quantification Methods

Angelika Dewicki, Matthew Henkel, Norie Sugitani, Alexander Applegate, and Brian Campfield

Corresponding Author(s): Brian Campfield, University of Pittsburgh School of Medicine

Review Timeline:

Submission Date:	June 13, 2025
Editorial Decision:	September 29, 2025
Revision Received:	November 22, 2025
Accepted:	November 26, 2025

Editor: Monika Kumaraswamy

Reviewer(s): Disclosure of reviewer identity is with reference to reviewer comments included in decision letter(s). The following individuals involved in review of your submission have agreed to reveal their identity: Rafael M. Cantón (Reviewer #1)

Transaction Report:

DOI: <https://doi.org/10.1128/spectrum.01807-25>

Re: Spectrum01807-25 (Performance of Time-Lapsed Turbidimetry and Agar Plating as Bacterial Quantification Methods)

Dear Dr. Brian T Campfield:

Thank you for submitting your manuscript entitled "Performance of Time-Lapsed Turbidimetry and Agar Plating as Bacterial Quantification Methods" to ASM Microbiology Spectrum. We have now received the reviewers' comments, and I am pleased to inform you that the overall feedback is positive.

However, the reviewers have also raised some concerns and suggestions that, if addressed, will significantly strengthen your manuscript. We invite you to revise your manuscript in light of these comments and submit the revised version for further consideration.

Please note that while a revision is encouraged, it does not guarantee acceptance of the manuscript. The revised version will be carefully evaluated by the reviewers and the editorial team.

Below you will find my comments, instructions from the Spectrum editorial office, and the reviewer comments.

Revision Guidelines

Sincerely,
Monika Kumaraswamy
Editor
Microbiology Spectrum

Reviewer #1 (Comments for the Author):

Authors compared in the manuscript two bacterial quantification methods: 1) traditional agar plating (CFU) and 2) liquid broth turbidimetry with *P. aeruginosa*, *K. pneumoniae* and *S. aureus*. Moreover, data were confronted with an in vivo murine model of pulmonary infection. Interestingly, results demonstrate that turbidimetry method provides comparable results with higher sensitivity, especially when low bacterial densities are tested. The study is well designed and presented. It is easy to follow.

Major comments

The manuscript would benefit from better positioning its novelty in relation to prior studies in the Introduction. Highlight that an in vivo models have been performed.

In the abstract there is a very categorical sentence (lines 16-17) that comparison of colony forming units and turbidity has not been performed. Are the authors completely sure of this? Which type of literature review they did? Even some references of this comparison are included (reference 3)

To perform bacterial counts, 10 mL were plated (line 94). Normally, 100 mL is done. This can affect to the limit of detection on bacterial counts. It needs to be discussed. This can also explain differences of both methods regarding limit of detection (LOD). Moreover, the authors state improved sensitivity with turbidimetry but do not numerically define the lower limit of detection (LOD) for either method. Adding quantified LOD values for all strains would improve clarity and replicability. This data can be included in the result section.

Were colony counts and turbidity performed in parallel using the same inoculum? Clarify for both in vitro and in vivo experiments.

Microorganism can be shortened one first mention has been performed (e.g. *K. pneumoniae*)

The manuscript uses "lag time" reduction as a surrogate for higher bacterial load in turbidimetry. This should be more explicitly justified in the Discussion. Include reference if possible.

Specific comments

Line 15. Change "access" to "assess" (typo).

Lines 111-114. Can the authors clarify how the Gen5 software algorithm identifies the transition from lag to exponential growth?

Lines 132-133. Were homogenates filtered or centrifuged to remove tissue debris prior to dilutions?

Line 304. Anaerobes and *S. pneumoniae*. Do the authors make an attempt with these microorganisms? Were limitations observed in both bacterial counts and turbidity?

Lines 311-313. Consider rephrasing to emphasize that this method may be preferable when subtle differences in bacterial load must be discerned

Reviewer #3 (Comments for the Author):

This is a head-to-head study comparing agar plating and liquid broth turbidimetry in bacterial quantification. The authors used 3 different bacterial isolates and 2 mouse models to study the correlation of the two methods along with the precision. They found that the methods correlated closely but liquid broth turbidimetry had greater precision and was able to detect growth at lower bacterial density than agar plating. The findings pose potential uses in research where bacterial quantification informs studies on infection progression or response to therapy, particularly where lower concentrations of bacteria are of interest. I found this to be a convincing and methodologically sound study that could inform future practice, particularly in the research setting. Clinical application may be limited given the widespread use of CFU in practice, but could be possible. The major critique of this study is that there were some typographical errors that affected outcome reporting (see detailed comments below) and there were references to data not shown from which conclusions were drawn. I would suggest that if the authors are going to draw conclusions, then they should include the data in the manuscript for the audience's independent interpretation.

Minor Comments

Introduction

Lines 55-57 "...*K. pneumoniae*, *P. aeruginosa*, and *S. aureus* are common MDR pathogens..." Consider re-wording. Not all *K. pneumoniae*, *P. aeruginosa* and *S. aureus* are MDR, although all are common pathogens.

Lines 64-65 Please characterize the strains of *K. pneumoniae*, *P. aeruginosa*, and *S. aureus*; were these drug resistant strains? Characterize here or in the methods. If not drug-resistant strains, then would recommend omitting reference to drug resistance as a point of interest of this study.

Lines 71-73 "This comparison identified..." This finding belongs in the discussion/conclusion, not the introduction.

Methods

Line 90 Could the authors comment on the choice of LB agar (as opposed to sheep blood agar which is more common in clinical micro lab setting). Presumably the choice was made given composition similarity to TSB to control for nutrient composition, etc. But sheep blood agar is more commonly used in the clinical setting; thus warrants some explanation.

Results

Line 194-195 "...extended detection to a 1:10,00,000 dilution" Please clarify whether this should be 1,000,000 or 10,000,000.

Comma placement and number of zeroes do not correlate thus obfuscating the interpretation.

Line 201 1:100,00 dilutions. Same issue as above.

Lines 226-232 "Tlr4, a critical component of innate immune function, is a canonical receptor for pathogen-associated molecular patterns...in vivo host defense model was assessed." Consider moving to introduction or even methods rather than results.

Lines 236-243 I found this paragraph hard to follow. The authors first compare spleen tissue growth from knock out vs WT on agar. Then lung tissue growth from knock out vs. WT in broth. Then back to spleen tissue in knockout vs. WT in broth. This section could be revised to compare: spleen on agar vs spleen in broth and lung on agar vs lung in broth, stratified by knock out and WT. The comparison of WT to knock out seems less relevant since that is not the question being considered.

Discussion

Line 270 Suggest including that detection of bacterial growth at lower concentrations is also valuable in the study of treatment response. I believe the authors mention this later as well.

Line 272 Authors note that liquid broth turbidimetry is more precise than agar and posit that this may be due to the "inherent manual handling variability associated with agar plating"; they may want to describe this process in the introduction to better make their point that liquid broth turbidimetry is less manual than agar plating.

Line 274 The authors posit that broth turbidimetry may reduce time needed to perform the assay but 'time' is not included in their results. Although they state "data not shown", I think they should not comment on time unless they introduce 'time' as a variable and include some results earlier in the manuscript in order to allow the audience to assess this as a conclusion. If not, then this conclusion should be omitted as not supported by data in the study.

Line 305 The authors introduce limitations regarding anaerobic and fastidious organisms, but they did not include these organisms in the methods or the results. As in my last comment, "data not shown" is not sufficient to cover this discussion, please include the data on these organisms in the methods and the results.

Line 307 Authors could consider mentioning a limitation to using broth turbidimetry in clinical practice. E.g. CFU is so entrenched in the practice of Infectious Disease medicine that reporting of OD would require a significant learning curve/practice change.

Reviewer #4 (Comments for the Author):

The authors have submitted a well-written manuscript describing their work toward a potential methodological advancement that has broad implications of interest to the readers of this journal. The use of multiple organisms in both in vitro and in vivo models helps to further demonstrate the potential of this method to answer questions critical to advancement of various fields. Addressing the following comments would help the general audience appropriately interpret some of the results and conclusions made by the authors:

Based on lines 88 and 99 describing different solutions used for dilutions, it appears that independent overnight inoculations and subsequent dilutions were made to test the two methods for in vitro comparison. A single source of each dilution would ideally be used for the two methods to make appropriate conclusions about limit of detection and precision. If this is not technically possible, it should be addressed in the conclusions.

Line 283: Data or references to literature describing the significance of the bacterial load changes detected within the lung of the mouse model would help support the conclusion that the statistical significance observed by turbidimetry is biologically relevant.

Figure 1: indicating the expected CFU/mL at each dilution based on the starting concentration would help provide context to the limit of detection data.

Line 307: a brief comment on differences in expense between the assays may be of interest to the general audience interested in this approach.

Re: Manuscript # Spectrum01807-25 **Performance of Time-Lapsed Turbidimetry and Agar Plating as Bacterial Quantification Methods**

Dear Dr. Kumaraswamy and Reviewers,

Thank you for the thoughtful critiques. We appreciate the opportunity to respond. Our responses follow the individual Reviewer comments below, and changes from the initial submission are in red font in the “Marked-Up Manuscript”.

Reviewer comments:

Reviewer #1

The manuscript would benefit from better positioning its novelty in relation to prior studies in the Introduction. Highlight that *in vivo* models have been performed.

Response: The Introduction has been edited to more clearly articulate the study’s contribution by defining the gap it addresses and highlighting the *in vivo* component. We have clarified that while individual calibration studies exist using these two methods, a comprehensive, systematic evaluation across a broad panel of clinically relevant pathogens has been lacking. Crucially, our novelty lies in extending this systematic comparison to a clinically relevant murine model of pulmonary infection.

In the abstract there is a very categorical sentence (lines 16-17) that comparison of colony forming units and turbidity has not been performed. Are the authors completely sure of this? Which type of literature review they did? Even some references of this comparison are included (reference 3)

Response: The original phrasing of the statement was incorrect, and we have revised the abstract for better precision. Strain-specific OD vs CFU calibration studies are common. However, our literature assessment confirmed the absence of a single, systematic study directly comparing the quantification performance of both methods across a panel of clinically relevant pathogens. Our revised abstract clarifies that our study fills this specific and important gap by providing a direct comparison under one standardized set of conditions.

To perform bacterial counts, 10 mL were plated (line 94). Normally, 100 mL is done. This can affect to the limit of detection on bacterial counts. It needs to be discussed. This can also explain differences of both methods regarding limit of detection (LOD)

Response: We confirm that the 10-microliter volume is our established laboratory protocol, chosen to minimize technical variation across the strains in this comparative study. We acknowledge that this choice raises the calculated limit of detection for CFU counts. This trade-off was felt to be acceptable because our primary focus in designing this study was on the higher-density bacterial samples. This justification and the discussion of its impact on the LOD comparison have been added to the discussion section of the revised manuscript.

Moreover, the authors state improved sensitivity with turbidimetry but do not numerically define the lower limit of detection (LOD) for either method. Adding quantified LOD values for all strains would improve clarity and replicability. This data can be included in the result section.

Response: We have included these values to enhance the clarity within the Results section for the *in vitro* experiments. We have numerically defined the LOD for both agar plating and liquid broth turbidimetry.

Were colony counts and turbidity performed in parallel using the same inoculum? Clarify for both *in vitro* and *in vivo* experiments

Response: In all experiments, both *in vitro* and *in vivo* colony counts and turbidity measurements were performed in parallel, utilizing aliquots from the same homogenized sample or culture. This step ensured that our colony counts and turbidimetry data are directly comparable, accurately reflecting the exact same microbial population. This procedure has been explicitly clarified and detailed within the materials and methods section of the revised manuscript.

Microorganism can be shortened one first mention has been performed (e.g. *K. pneumoniae*)

Response: This comment has been addressed in the revised manuscript to shorten the microorganism names after the first mention.

The manuscript uses "lag time" reduction as a surrogate for higher bacterial load in turbidimetry. This should be more explicitly justified in the Discussion. Include reference if possible.

Response: We have explicitly justified our interpretation of lag time reduction as a surrogate measure for higher initial bacterial load in the liquid broth turbidimetry assay. This interpretation is based on a foundational principle of microbial growth kinetics, where the time required for a culture to reach a detectable OD threshold or lag time is inversely proportional to the number of starting viable bacterial cells. We have revised the Discussion section to clearly state this principle and its application to our *in vivo* data, ensuring complete methodological transparency. References have been added that convey this principle.

Lines 111-114. Can the authors clarify how the Gen5 software algorithm identifies the transition from lag to exponential growth?

Response: Lag time was calculated mathematically using the data reduction features of the Gen5 software. This does not rely on simple visual inspection for phase transition. The Gen5 software determined the lag phase duration by fitting the kinetic OD₆₀₀ data to a standard non-linear regression growth model. The software then used this fitted curve to calculate the maximum specific growth rate and defined the lag time as the time-axis intercept. This precise computational methodology has been added to the materials and methods section of the revised manuscript.

Lines 132-133. Were homogenates filtered or centrifuged to remove tissue debris prior to dilutions?

Response: Lung and spleen homogenates were not filtered or centrifuged prior to serial dilution and plating. Introducing filtration or centrifugation carries a significant risk of losing or altering the bacterial load due to adherence to filters or pelleting inefficiencies. Since our goal was the accurate and complete recovery of viable organisms, we prioritized minimizing these processing artifacts. The subsequent serial dilutions rapidly reduced the concentration of host tissue debris, ensuring it did not interfere with the final CFU plating and turbidity.

Line 304. Anaerobes and *S. pneumoniae*. Do the authors make an attempt with these microorganisms? Were limitations observed in both bacterial counts and turbidity?

Response: Anaerobes were not utilized within this study due to their requirement for a specialized plate reader capable of supporting strict anaerobic conditions for the turbidimetry analysis. This was added to the Discussion section as a limitation.

Lines 311-313. Consider rephrasing to emphasize that this method may be preferable when subtle differences in bacterial load must be discerned

Response: We have revised the statement in lines 311-313 to emphasize that the enhanced sensitivity and reliability of the turbidimetry method allow for it to discern subtle, yet significant, differences in bacterial load and growth kinetics. This is achieved because the method's automated, continuous monitoring process provides the precision and low technical variability for a more refined evaluation needed for high-throughput assays.

Reviewer #3

Lines 55-57 "...*K. pneumoniae*, *P. aeruginosa*, and *S. aureus* are common MDR pathogens..." Consider re-wording. Not all *K. pneumoniae*, *P. aeruginosa* and *S. aureus* are MDR, although all are common pathogens.

Response: The intention of the original text was to highlight the profound clinical relevance of infection by these organisms, which are often MDR. We have revised the Introduction to remove the categorical statement and accurately reflect the clinical impact. This revision establishes that these species are studied due to their clinical importance and association with the global MDR threat.

Lines 64-65 Please characterize the strains of *K. pneumoniae*, *P. aeruginosa*, and *S. aureus*; were these drug resistant strains? Characterize here or in the methods. If not drug-resistant strains, then would recommend omitting reference to drug resistance as a point of interest of this study.

Response: The strains were selected for their clinical relevance and frequent use in pulmonary infection models, not specifically for being multi-drug resistant. We have edited the categorical

statement implying all strains are MDR from the Introduction. Lines 71-73 "This comparison identified..." This finding belongs in the discussion/conclusion, not the introduction.

Response: This statement was removed from the Introduction and was supplemented into the Discussion section.

Methods

Line 90 Could the authors comment on the choice of LB agar (as opposed to sheep blood agar which is more common in clinical micro lab setting). Presumably the choice was made given composition similarity to TSB to control for nutrient composition, etc. But sheep blood agar is more commonly used in the clinical setting; thus warrants some explanation.

Response: Luria-Bertani (LB) agar was used instead of the sheep blood agar (common clinical media) because LB is a standard, rich, non-selective medium consistently employed in microbiological research settings. Our primary objective was to understand the fundamental performance biases and dynamic range difference between turbidimetry and agar plating under controlled research conditions, not to mimic a clinical diagnostic environment. Using a single, standardized research medium ensured consistent experimental conditions across our panel of pathogens, which is essential for a systematic and unbiased comparison. This justification for the choice of LB agar has been added to the materials and methods section of the revised manuscript.

Results

Line 194-195 "...extended detection to a 1:10,00,000 dilution" Please clarify whether this should be 1,000,000 or 10,000,000. Comma placement and number of zeroes do not correlate thus obfuscating the interpretation.

Response: The change to the correct value was made in the revised manuscript to accurately reflect the experimental results.

Line 201 1:100,00 dilutions. Same issue as above.

Response: The change to the correct value was made in the revised manuscript to accurately reflect the experimental results.

Lines 226-232 "Tlr4, a critical component of innate immune function, is a canonical receptor for pathogen-associated molecular patterns...in vivo host defense model was assessed." Consider moving to introduction or even methods rather than results.

Response: This statement has been moved from the results section to the methods section following the introduction of the mouse model, and mention made in the Introduction, of the revised manuscript.

Lines 236-243 I found this paragraph hard to follow. The authors first compare spleen tissue growth from knock out vs WT on agar. Then lung tissue growth from knock out vs. WT in broth. Then back to spleen tissue in knockout vs. WT in broth. This section could be revised to compare: spleen on agar vs spleen in broth and lung on agar vs lung in broth, stratified by knock out and WT. The comparison of WT to knock out seems less relevant since that is not the question being considered.

Response: The main goal of this paragraph was to compare the performance of the two quantification methods within the *in vivo* model. We have revised the paragraph to prioritize the methodological comparison. The revised structure now presents the data by explicitly comparing spleen CFU vs spleen lag time and lung CFU vs lung lag time across both mouse genotypes.

Discussion

Line 270 Suggest including that detection of bacterial growth at lower concentrations is also valuable in the study of treatment response. I believe the authors mention this later as well.

Response: The enhanced sensitivity of the turbidimetry method is highly valuable not only in studying early infection stages but also in the context of treatment efficacy. We have revised the relevant statement in the Discussion section to incorporate this application.

Line 272 Authors note that liquid broth turbidimetry is more precise than agar and posit that this may be due to the "inherent manual handling variability associated with agar plating"; they may want to describe this process in the introduction to better make their point that liquid broth turbidimetry is less manual than agar plating.

Response: We have revised the Introduction to include a brief, comparative description that explicitly details the sources of technical variability in the traditional method of agar plating. While agar plating remains the standard, its execution involves numerous manual steps which contribute to inherent technical variability.

Line 274 The authors posit that broth turbidimetry may reduce time needed to perform the assay but 'time' is not included in their results. Although they state "data not shown", I think they should not comment on time unless they introduce 'time' as a variable and include some results earlier in the manuscript in order to allow the audience to assess this as a conclusion. If not, then this conclusion should be omitted as not supported by data in the study.

Response: The comment on broth turbidimetry reducing time needed to perform the assay was removed as it was not introduced as a variable within the results section.

Line 305 The authors introduce limitations regarding anaerobic and fastidious organisms, but they did not include these organisms in the methods or the results. As in my last comment, "data not shown" is not sufficient to cover this discussion, please include the data on these organisms in the methods and the results.

Response: The comment regarding anaerobic and fastidious organisms has been edited to address this limitation. These organisms were not thoroughly studied, so no data were added.

Line 307 Authors could consider mentioning a limitation to using broth turbidimetry in clinical practice. E.g. CFU is so entrenched in the practice of Infectious Disease medicine that reporting of OD would require a significant learning curve/practice change.

Response: We agree that the entrenched status of CFU reporting in infectious disease medicine represents a significant barrier to implementation. We have added a clarifying statement to the Discussion section of the revised manuscript to address this limitation.

Reviewer #4

Based on lines 88 and 99 describing different solutions used for dilutions, it appears that independent overnight inoculations and subsequent dilutions were made to test the two methods for in vitro comparison. A single source of each dilution would ideally be used for the two methods to make appropriate conclusions about limit of detection and precision. If this is not technically possible, it should be addressed in the conclusions.

Response: This comment was previously addressed for reviewer #1. Please refer to the response above

Line 283: Data or references to literature describing the significance of the bacterial load changes detected within the lung of the mouse model would help support the conclusion that the statistical significance observed by turbidimetry is biologically relevant.

Response: We have revised the Discussion section to explicitly support the conclusion that the improved statistical significance is biologically relevant with references. Bacterial burden changes in the lung are a direct measure of an immune response defect and increased bacterial load.

Figure 1: indicating the expected CFU/mL at each dilution based on the starting concentration would help provide context to the limit of detection data.

Response: We have revised the legend of Figure 1 to include the necessary quantitative context by indicating that the dilutions were expected to decrease by tenfold from the initial 1:1 dilution

factor. This addition provides the essential quantitative framework to evaluate the performance, dynamic range, and LOD differences between the CFU and OD quantification.

Line 307: a brief comment on differences in expense between the assays may be of interest to the general audience interested in this approach.

Response: The economic and practical considerations are relevant for labs adopting the high-throughput approach. We have added a clarifying statement to the Discussion section that addresses this aspect, discussing the initial investment of a plate reader.

Re: Spectrum01807-25R1 (Performance of Time-Lapsed Turbidimetry and Agar Plating as Bacterial Quantification Methods)

Dear Dr. Brian T Campfield:

Your manuscript has been accepted, and I am forwarding it to the ASM production staff for publication. Your paper will first be checked to make sure all elements meet the technical requirements. ASM staff will contact you if anything needs to be revised before copyediting and production can begin. Otherwise, you will be notified when your proofs are ready to be viewed.

Sincerely,
Monika Kumaraswamy, MD, D(ABMM)
Editor
Microbiology Spectrum